# The Inversion of HY-1C-COCTS Ocean Color Remote Sensing Products from High-Latitude Seas

**Hao Li** [1,2], **Xianqiang He** [1,3,*], **Jing Ding** [4], **Yan Bai** [1], **Difeng Wang** [1], **Fang Gong** [1] and **Teng Li** [1]

1   State Key Laboratory of Satellite Ocean Environment Dynamics, Second Institute of Oceanography, Ministry of Natural Resources, Hangzhou 310012, China
2   The Fourth Institute of Oceanography, Ministry of Natural Resources, Beihai 536000, China
3   Donghai Laboratory, Zhoushan 316021, China
4   National Satellite Ocean Application Service, Beijing 100082, China
*   Correspondence: hexianqiang@sio.org.cn

**Abstract:** China's first operational ocean color satellite sensor, named the Chinese Ocean Color and Temperature Scanner (HY-1C-COCTS), was launched in September 2018 and began to provide operational data in June 2019. However, as a polar orbiting ocean color satellite sensor, HY-1C-COCTS would inevitably encounter regions impacted by large solar zenith angles when observing the high-latitude seas, especially during the winter. The current atmospheric correction algorithm used by ocean color satellite data processing software cannot effectively process observation data with solar zenith angles greater than $70°$. This results in a serious lack of effective ocean color product data from high-latitude seas in winter. To solve this problem, this study developed an atmospheric correction algorithm based on a neural network model for use with HY-1C-COCTS data. The new algorithm used HY-1C-COCTS satellite data collected from latitudes greater than $50°N$ and between April 2020 and April 2021 to establish a direct relationship between the total radiance received by the satellite and the remote sensing reflectance products. The evaluation using the test dataset shows that the inversion accuracy of the new algorithm is relatively high under different solar zenith angles and different HY-1C-COCTS bands (the relative deviation is 3.37%, 7.05%, 5.10%, 5.29%, and 10.06% at 412 nm, 443 nm, 490 nm, 520 nm, and 565 nm, respectively; the relative deviation is 1.07% when the solar zenith angle is large ($70~90°$)). Cross comparison with MODIS Aqua satellite products shows that the inversion results are consistent. After verifying the accuracy and stability of the algorithm, we reconstructed the remote sensing reflectance dataset from the Arctic Ocean and surrounding high-latitude seas (latitude greater than $50°N$) and successfully retrieved chlorophyll-a data and information on other marine ecological parameters.

**Keywords:** ocean color; HY-1C-COCTS; atmospheric correction; polar zone; diurnal change; large solar zenith angle

## 1. Introduction

China's first operational ocean color satellite sensor, named the Chinese Ocean Color and Temperature Scanner (HY-1C-COCTS), was launched in September 2018 and began to provide operational data in June 2019. The HY-1C satellite is equipped with an ocean color and temperature scanner, ultraviolet imager, and coastal zone imager, which have important applications in typhoon, sea ice, marine oil spill, cyanobacteria, *Enteromorpha prolifera*, and forest fire monitoring [1,2]. For example, based on HY-1C data, Lu et al. (2019) successfully identified an offshore oil spill event near Dongsha Island in the South China Sea on 20 February 2019 [3]. Using the data obtained by the coastal zone imager on HY-1C, Shen et al. (2020) successfully distinguished the oil slick and oil spill emulsion formed by the oil spill [4]. However, as a polar orbiting ocean color satellite, HY-1C would inevitably encounter environments impacted by large solar zenith angles (SZA) when observing

the high-latitude seas. The current standard atmospheric correction algorithm used by ocean color satellite data processing software (such as ACOLITE) cannot effectively process observation data with SZAs greater than 70° [5–7]. Figure 1 shows the observation data from HY-1C-COCTS for Baffin Bay on 5 October 2020, where Figure 1a is a true color image, Figure 1b is a remote sensing reflectance (Rrs) product at the 443 nm band, and Figure 1c is the corresponding SZA. It can be seen that although there are some cloudless areas (dark blue in the figure), due to the failure of the atmospheric correction algorithm under a large SZA, the amount of effective data of the Rrs products is very small. In observational environments impacted by a large SZAs, the influence of Earth's curvature increases, the Rayleigh scattering lookup table used by the traditional atmospheric correction algorithm has errors, aerosol properties are complex and changeable, and aerosol radiance is difficult to accurately estimate [8,9]. In recent years, Li et al. (2019) quantitatively analyzed the remote sensing detectability of the three ocean color components (chlorophyll, suspended solids, and CDOM) from environments impacted by large SZAs based on the vector radiative transfer model for coupled ocean–atmosphere system and proved that even if the SZA reaches 80°, minor changes in ocean color components such as suspended solids can be detected [10,11]. After determining that the satellite is capable of detection in environments impacted by large SZAs, Li et al. (2020) established the atmospheric correction method for morning and evening observation data from the geostationary ocean color satellite sensor GOCI [5]. This technique demonstrated the effective processing of the satellite morning and evening observation data from ocean water. On this basis, and in view of the serious lack of effective HY-1C-COCTS ocean color products from high-latitude seas, this study developed an atmospheric correction algorithm based on a neural network model that can be applied to HY-1C-COCTS data collected during large SZAs. The objective of the model is to achieve effective recovery of HY-1C-COCTS ocean color remote sensing products in the high-latitude seas.

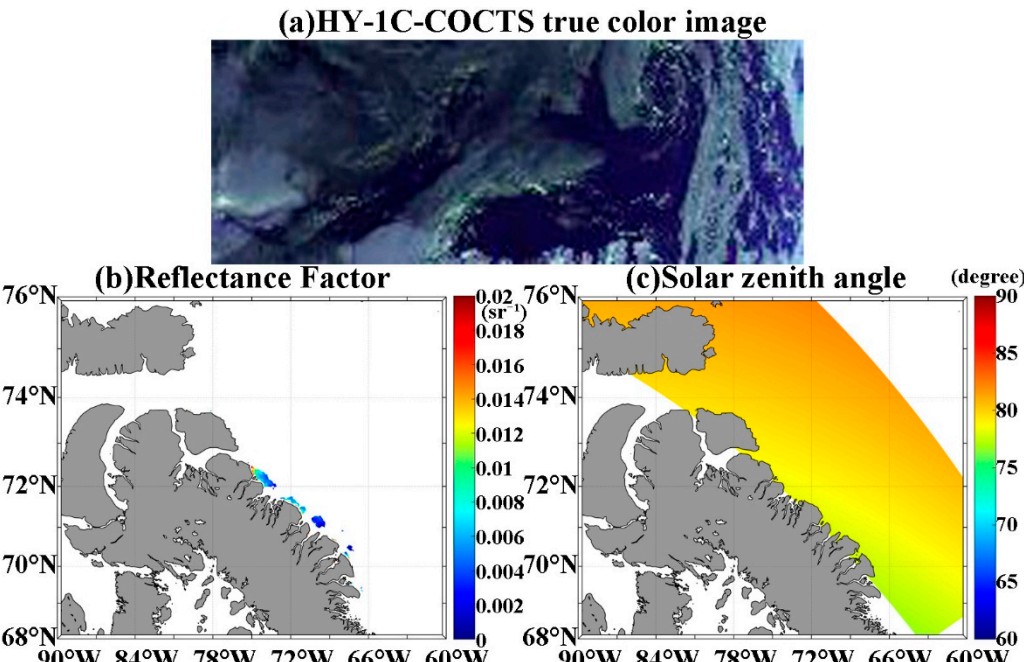

**Figure 1.** Observation data of HY-1C-COCTS in Baffin Bay on 5 October 2020: (**a**) true color image; (**b**) Rrs in the 443 nm band; (**c**) SZA.

## 2. Data and Methods

### 2.1. Satellite Data

The HY-1C-COCTS has six visible light bands and two near-infrared bands, i.e., 412 nm, 443 nm, 490 nm, 520 nm, 565 nm, 670 nm, 750 nm, and 865 nm, with the spa-

tial resolution of 1.1 km. In this study, we obtained Level-1B data of HY-1C-COCTS with latitudes higher than 50°N from April 2020 to April 2021 from the National Satellite Ocean Application Service to build an applicable atmospheric correction algorithm for the processing of the large SZA observation data. To verify the accuracy of the established atmospheric correction algorithm, we also obtained certain MODIS/Aqua Level-2 RF products from NASA for cross-comparison and verification between satellites (Download link: https://oceancolor.gsfc.nasa.gov/, accessed on 2 December 2020) [12,13]. The MODIS Level-2 products released by NASA are from the Level-1 products processed by the near-infrared (NIR) iterative atmospheric correction algorithm. The algorithm first assumes that the leaving-water reflectance of the near-infrared band is 0 and then extrapolates the aerosol reflectance of the near-infrared band to the visible band to calculate the leaving-water reflectance of each band. If the calculated leaving-water reflectance of the near-infrared band changes little compared with the last iteration, the iteration is terminated to obtain the final result. The accuracy of MODIS/Aqua Level-2 Rrs products had already been validated by the NOMAD dataset with mean absolute errors of $0.00104 \, \text{sr}^{-1}$, $0.00083 \, \text{sr}^{-1}$, $0.00097 \, \text{sr}^{-1}$, $0.00089 \, \text{sr}^{-1}$, $0.00094 \, \text{sr}^{-1}$, $0.00109 \, \text{sr}^{-1}$, $0.00034 \, \text{sr}^{-1}$, and $0.00032 \, \text{sr}^{-1}$ for the 412 nm, 443 nm, 488 nm, 531 nm, 547 nm, 555 nm, 667 nm, and 678 nm, respectively [14].

### 2.2. Neural Network Atmospheric Correction Model

Atmospheric correction can be regarded as a nonlinear function approximation of the input spectrum, and the use of machine learning methods can efficiently extract target information under complex conditions [15,16]. In recent years, many studies have adopted machine learning methods to the atmospheric correction processes for satellite ocean color sensors such as GOCI, MERIS, and Sentinel-2. For example, Fan et al. (2017, 2020) proposed a multilayer neural network atmospheric correction algorithm for offshore turbid water algorithm based on the simulated dataset by the radiative transfer model AccuRT to train the neural network, which can effectively process the nearshore ocean color data of the polar orbiting satellite sensor MODIS [17,18]. Tian et al. (2014) developed an atmospheric correction algorithm based on the neural network model, which was applicable to GOCI and could effectively process the observation data of turbid water bodies in the Bohai Sea [19,20]. Shen et al. (2019) evaluated the applicability of the neural network model in Case II water and compared it with the traditional atmospheric correction algorithm, and they found that the neural network model had advantages in data processing efficiency and accuracy [21,22]. Chen et al. successively constructed algorithms for total suspended solids concentration, water absorption and scattering coefficient, diffuse attenuation coefficient, and dissolved organic matter absorption coefficient based on neural network method. Compared with traditional empirical algorithms, the neural network method was more efficient and accurate [23–25].

The accuracy of the atmospheric correction model based on the neural network method depends on the representativeness of the training dataset. Due to the difficulty of an accurate radiative transfer simulation under large SZAs, the trained neural network model based on the simulation dataset will also have problems under large SZA conditions. In this study, the processed HY-1C-COCTS Level-1 and Level-2 products were used to build the training dataset, and then, the neural network model was built based on it.

#### 2.2.1. Construction of the Training Dataset

The total signal from the top of the atmosphere that is received by the ocean color satellite sensor can be expressed as

$$L_t(\lambda) = L_r(\lambda) + L_a(\lambda) + L_{ra}(\lambda) + T(\lambda) * L_g(\lambda) + t(\lambda) * L_{wc}(\lambda) + T(\lambda) * L_w(\lambda) \quad (1)$$

where $L_r(\lambda)$ is the Rayleigh scattering radiance, $L_a(\lambda)$ is the aerosol radiance, $L_g(\lambda)$ is the sunlight radiance, $L_{wc}(\lambda)$ is the radiance caused by white caps on the sea surface, $L_w(\lambda)$ is the leaving-water radiance, and $t(\lambda)$ and $T(\lambda)$ are the diffuse and direct atmospheric transmittances, respectively. $L_{ra}(\lambda)$ is the radiance contributed by the interaction between

Rayleigh scattering and aerosol scattering, and it is the result of photons scattering by both the atmospheric molecules and aerosols. For the single scattering case, $L_{ra}(\lambda)$ is 0.

The influence of sun glint can be masked by the sun-satellite observation geometry. The contribution of white cap radiance to the total radiance is negligible when the sea surface wind speed is not extremely high. According to the information of wind speed, pressure, and band, the Rayleigh scattering radiance can be accurately calculated by using the predefined Rayleigh scattering lookup table. However, the Rayleigh scattering lookup table generated by the traditional vector radiative transfer model using the assumption of plane atmospheric stratification does not consider the effect of the curvature of the Earth and has a large error and poor accuracy when the SZA is large. In this study, the vector radiative transfer model for coupled ocean–atmosphere system named PCOART-SA [26,27], which considers the effect of the curvature of the Earth, was used to generate the Rayleigh scattering lookup table for the HY-1C-COCTS.

In the high-latitude seas, HY-1C-COCTS has up to 14 repeated observations every day (daily observation times increase with increasing latitude), like the hourly observations of geostationary ocean color satellites. Therefore, we can link the Rayleigh-corrected total radiance at the dawn or dusk (high SZAs) and the remote sensing reflectance (Rrs) retrieved by standard atmospheric correction algorithm near noontime (low-to-moderate SZAs) to establish the training dataset. To establish such a training dataset, we selected HY-1C-COCTS images with cloud coverage less than 60% over the ocean and latitudes higher than 50°N from April 2020 to April 2021 and then processed them according to the following steps:

(1) First, all known quantities in the total signal received by the satellite are deducted, and the Rayleigh-corrected radiance is calculated by

$$L_{rc}(\lambda) = L_t(\lambda) - T(\lambda) * L_g(\lambda) - t(\lambda) * L_{wc}(\lambda) - L_r(\lambda) \tag{2}$$

Rayleigh-corrected radiance was used as the input data of the neural network model. Additionally, the SZA, observation zenith angle and relative azimuth were be used as the input data of the neural network model.

(2) The near-infrared iterative atmospheric correction algorithm was used to obtain the Rrs products of the noontime observations. The near-infrared iterative atmospheric correction algorithm uses two near-infrared bands from HY-1C-COCTS (750 nm, 865 nm) as reference bands to determine the aerosol type and optical thickness. According to the predefined aerosol scattering lookup table and atmospheric diffuse transmittance lookup table, the atmospheric path radiance of the visible light band was extrapolated [28]. Further, the following filtering criteria were used to screen high-quality Rrs data: (1) Extract the 3 × 3 pixel frame where the percentage of effective Rrs pixels are greater than 50% of those within the pixel frame (excluding land pixels), and conduct the next step; (2) Calculate the average and standard deviation (SDs) of the effective Rrs values in the 3 × 3 pixel frame. Discard the pixel whose Rrs value exceeds 1.5 times the standard deviation of the average value; (3) Recalculate the mean and standard deviation of the remaining effective pixels, and calculate the coefficient of spatial variation (CV). The pixel frames with coefficients of variation greater than 0.15 were discarded; (4) For the diurnal multiple observations, select two to four observations near noontime to check the time stability of Rrs, and calculate the time variation coefficient (CV). The pixel frame with a coefficient of variation less than 0.15 was selected.

Among these, the first three criteria ensure the consistency of the filtered data in the spatial range and avoid noise caused by atmospheric correction or stray clouds. The fourth criterion ensures the consistency of the selected data within the time range and avoids rapid changes in water bodies caused by blooms or strong currents.

(3) After filtering the high-quality Rrs data, the Rayleigh-corrected radiance, SZA, observation zenith angle, and relative azimuth data with the same position in the 3-hour

time window but large SZA (morning and evening hours) were used to match the noontime Rrs data. The size of the time window is the same as that proposed by the Marine Biological Processing Group for matching satellite and in situ data [29,30]. The distribution of the SZA in the final dataset is shown in Figure 2.

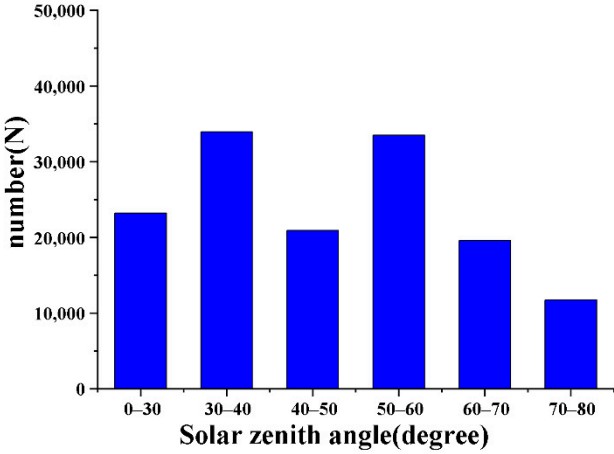

**Figure 2.** The distribution of the SZA in the neural network training dataset.

2.2.2. Building the HY-1C-COCTS Neural Network Atmospheric Correction Model

In this study, the Rrs products are obtained directly from the Rayleigh-corrected radiance retrieved from satellites using the neural network training method. Neural networks are a powerful tool for prediction, recognition, function approximation, and pattern classification [31,32]. Therefore, it is feasible to train a neural network to find the relationship between Rayleigh-corrected radiance and Rrs. The neural network model established includes an input layer, output layer, and hidden layer. The input layer contains the SZA, observation zenith angle, relative azimuth angle, and Rayleigh-corrected radiance of the HY-1C-COCTS.

When constructing neural networks, it is necessary to determine the optimal number of middle layers (or hidden layers) and neurons according to the input and output parameters, training samples, and function complexity. Through the comparison between the single-layer neural network model and the multilayer (three hidden layers) neural network model, it was found that the single-layer neural network has the same accuracy as the multilayer model, but the training data require less time. Therefore, we chose a single-layer neural network as the simulation tool. We also compared the effects of different numbers of neurons and finally decided to use nine neurons.

Specifically, this work establishes a neural network with a single hidden layer to determine the relationship between Rayleigh-corrected radiance and Rrs. The neural network model includes an input layer, output layer, and hidden layer, which contains nine neurons. There are nine elements in the input layer, including Rayleigh-corrected radiance, SZA, observation zenith angle, and relative azimuth of 6 HY-1C-COCTS bands (412 nm, 443 nm, 490 nm, 520 nm, 565 nm, and 670 nm). There are six elements in the output layer, which are the Rrs of six HY-1C-COCTS bands. The standardization process of the input and output parameters is automatically completed by the MATLAB neural network toolbox. The initial value of the neural network is randomly obtained by the system, and the value range is 0~1. The weight value and deviation value are updated iteratively during training. The training results are evaluated using the training target (i.e., the Rrs of the general SZA) and the root mean square error of the neural network output results. After the neural network is ready, the training dataset is used to train and test the model. The whole training dataset is divided into a model training dataset and a precision evaluation dataset, accounting for 70% and 30% of the total dataset, respectively. Finally, the above process was performed in MATLAB.

*2.3. Statistical Parameters*

Root mean square deviation (RMSE), absolute percentage deviation (APD), and relative percentage deviation (RPD) are used to evaluate the algorithm accuracy. The formulae of these statistical parameters are as follows:

$$\text{RMSE} = \sqrt{\frac{\sum_{i=1}^{N}(y_i - x_i)^2}{N}} \tag{3}$$

$$\text{APD} = 100\% * \frac{1}{N}\sum_{i=1}^{N}\frac{|y_i - x_i|}{x_i} \tag{4}$$

$$\text{RPD} = 100\% * \frac{1}{N}\sum_{i=1}^{N}\frac{y_i - x_i}{x_i} \tag{5}$$

## 3. Results

### 3.1. Model Verification

To verify the accuracy of the atmospheric correction model established for the HY-1C-COCTS large SZA observation data, the precision evaluation dataset was used. The precision evaluation dataset comes from another 30% of the training dataset. Figure 3 shows the inversion results from the established atmospheric correction algorithm in each HY-1C-COCTS band. The scatter plot shows that the inversion results from the neural network model in all HY-1C-COCTS bands are around the 1:1 line. In general, the retrieved Rrs values are consistent with the known values, and the correlation coefficients are greater than 0.91. The statistical parameters of the neural network atmospheric correction model in each band are shown in Table 1. The inversion results are more accurate in the shorter wavelength band. In the visible-light band, the absolute error percentage is within 11% (the APD values of 412 nm, 443 nm, 490 nm, 520 nm, and 565 nm are 3.37%, 7.05%, 5.10%, 5.29%, and 10.06%, respectively). These results show that the neural network atmospheric correction model can accurately learn from the training dataset.

**Table 1.** The statistical parameters obtained by the algorithm are verified using precision evaluation datasets for different bands.

| Band | RMSD ($sr^{-1}$) | APD (%) | RPD (%) |
|------|------------------|---------|---------|
| 412 nm | 0.00075 | 0.45% | 3.37% |
| 443 nm | 0.00079 | 1.15% | 7.05% |
| 490 nm | 0.00069 | 0.22% | 5.10% |
| 520 nm | 0.00070 | 0.14% | 5.29% |
| 565 nm | 0.00080 | −0.37% | 10.06% |
| 670 nm | 0.00071 | 26.12% | 48.68% |

Figure 4 shows the accuracy of the neural network atmospheric correction model based on the precision evaluation dataset at different SZA intervals. Under different SZAs, the inversion accuracy of the algorithm has no obvious difference, and the accuracy does not decrease with increasing SZA. It can be seen from the statistical parameters in Table 2 that RMSE, APD, and RPD have no significant difference under different SZAs. For example, under the general SZA observation environment (30~40°N), the RMSE, APD, and RPD are 0.00073 $sr^{-1}$, 0.70%, and 6.80%, respectively, while under the large SZA observation environment (70~90°N), RMSE, APD, and RPD are 0.00028 $sr^{-1}$, 0.44%, and 1.07%, respectively. Overall, the atmospheric correction algorithm established is applicable to observational environments impacted by large SZAs.

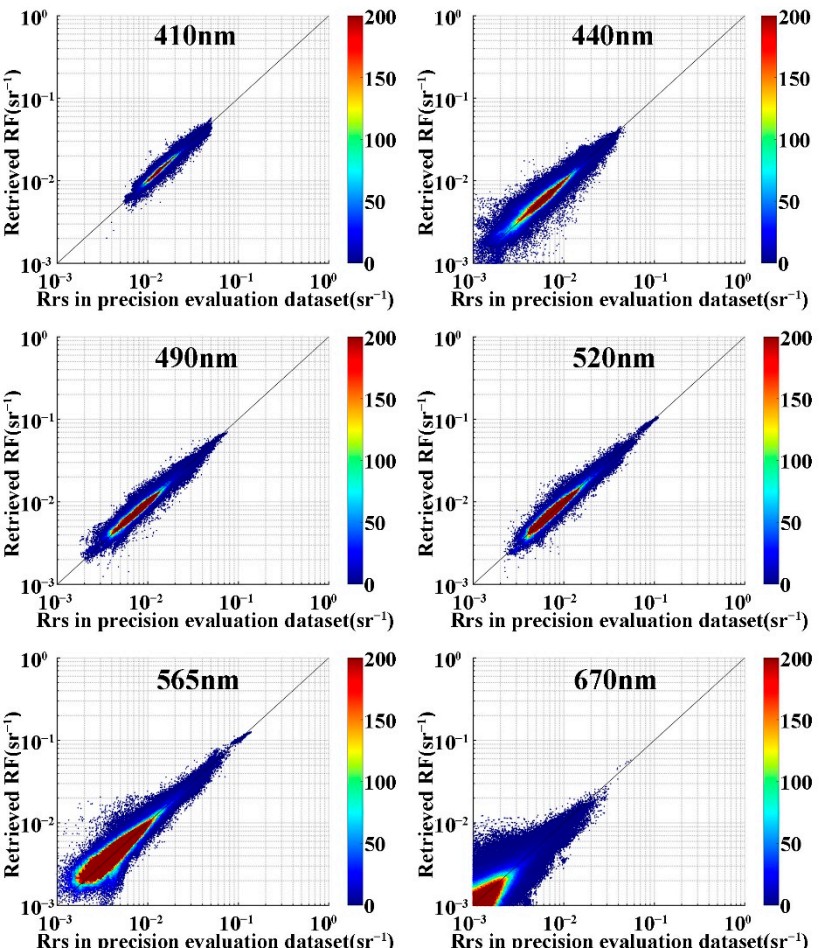

**Figure 3.** Inversion results of the neural network atmospheric correction algorithm in each HY-1C-COCTS band. The color bar represents the number of points.

**Table 2.** The statistical parameters obtained by the algorithm are verified using precision evaluation datasets for different SZAs.

| SZA | RMSD (sr$^{-1}$) | APD (%) | RPD (%) |
|-----|------------------|---------|---------|
| 0~30° | 0.00078 | 0.89% | 5.95% |
| 30~40° | 0.00073 | 0.70% | 6.80% |
| 40~50° | 0.00077 | 2.56% | 8.57% |
| 50~60° | 0.00091 | 1.44% | 8.85% |
| 60~70° | 0.00060 | 0.26% | 3.83% |
| 70~90° | 0.00028 | 0.44% | 1.07% |

### 3.2. Cross Validation of Satellite Products

To further evaluate the reliability of the established atmospheric correction model, we used the MODIS/Aqua product to cross-verify the model results. HY-1C-COCTS data were processed and compared using the neural network atmospheric correction model and the NIR iterative atmospheric correction algorithm. Figure 5 shows the comparison results in the eastern Pacific on 2 December 2020. When the latitude is low, and the SZA is small, the inversion results of the neural network atmospheric correction model are consistent with the results of the NIR iterative atmospheric correction algorithm, and the product distribution between different satellites is basically the same. Figure 6a shows a scatter plot comparing the results of the two algorithms. The scattered points are concentrated near the 1:1 line. The correlation coefficient between the neural network atmospheric correction model and the traditional NIR iterative atmospheric correction algorithm is 0.90, and the



relative deviation is 24%. Figure 6b shows the product cross-comparison results of the two satellites, and the statistical parameters show that the correlation coefficient is 0.65. Due to the difference in observation time and spatial resolution between the two satellites, the deviation is relatively large, but the scattered points are also concentrated around the 1:1 line. To further evaluate the stability of the model, we compared the monthly average data of MODIS/Aqua and HY-1C-COCTS in summertime, with the results shown in Figure 7. Comparing Figures 7a and 7b, it can seem that when the SZA is small, the results of the two satellites are similar, and their distributions and values are relatively close. The scattered points in different bands are also concentrated near the 1:1 line. In general, when processing satellite images acquired over a general observation environment, the inversion result of the atmospheric correction model established here is equivalent to that of the traditional atmospheric correction algorithm, and thus, the algorithm result is reliable.

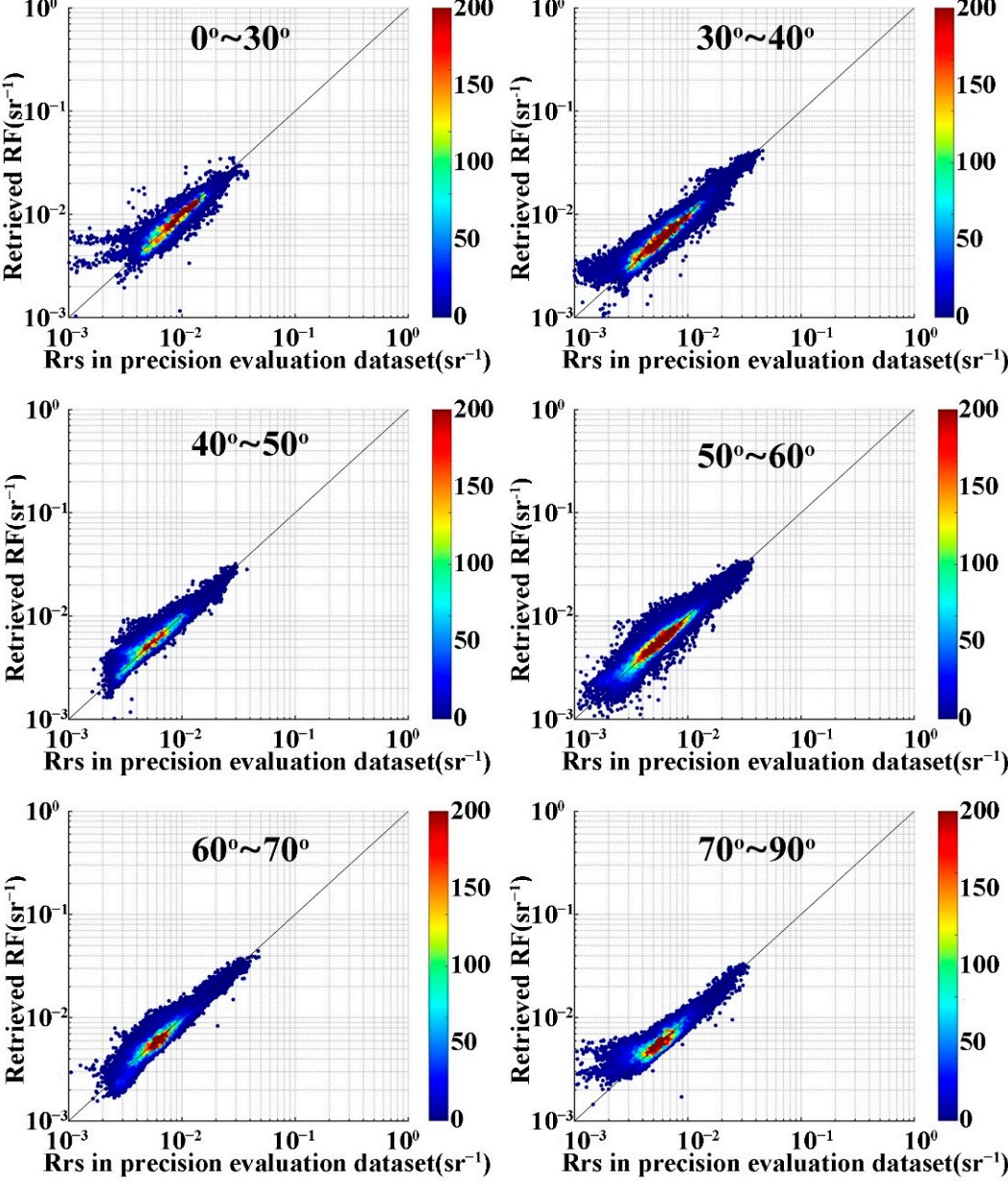

**Figure 4.** Inversion results of the neural network atmospheric correction algorithm in different SZAs. The color bar represents the number of points.

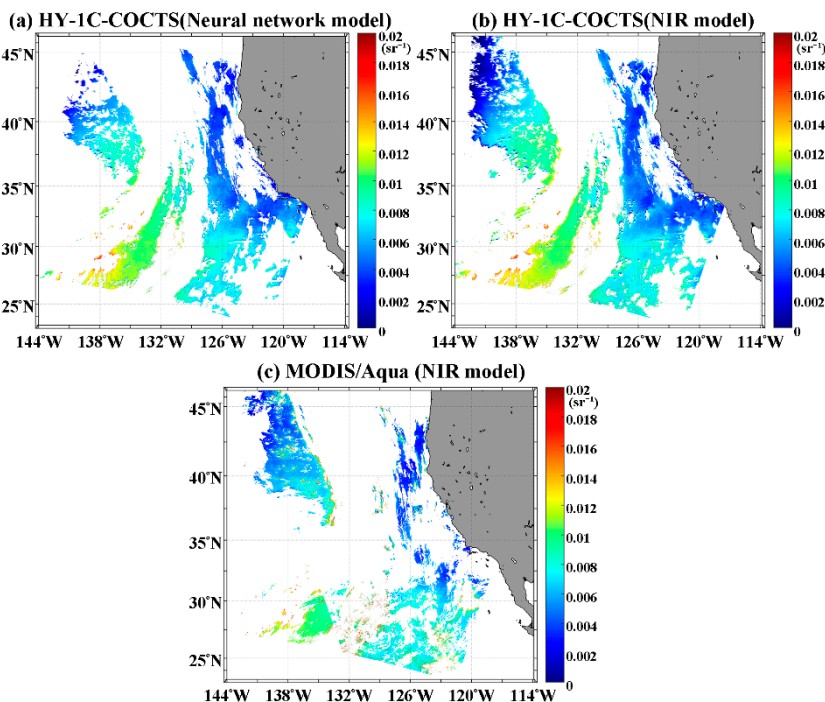

**Figure 5.** Satellite Rrs (443 nm) products: (**a**) HY-1C-COCTS products processed by the neural network atmospheric correction algorithm; (**b**) HY-1C-COCTS products processed by the NIR algorithm; (**c**) the product of MODIS/Aqua processed by the NIR algorithm.

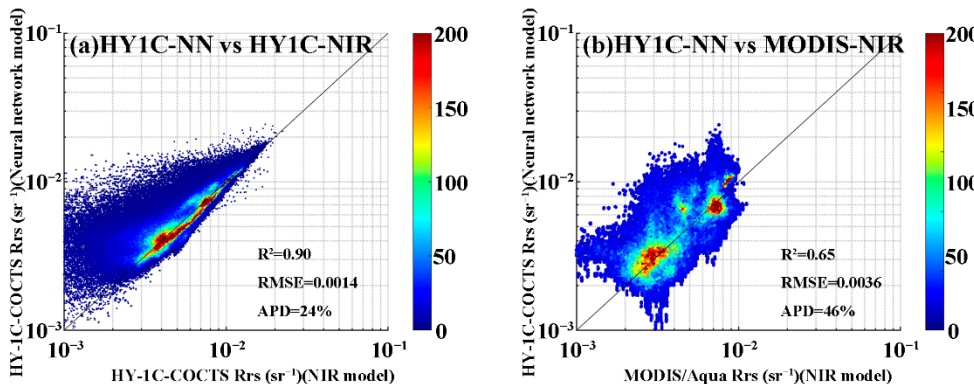

**Figure 6.** Comparison of the 443 nm Rrs products retrieved by the neural network atmospheric correction algorithm and the inversion results of the near-infrared iterative atmospheric correction algorithm. (**a**) Comparison of the results of two algorithms for processing HY-1C-COCTS. (**b**) Comparison of the results of two algorithms for processing HY-1C-COCTS and MODIS/Aqua data.

Figure 8 shows the 443 nm Rrs product from the HY-1C-COCTS satellite acquired on 5 October 2020 using the neural network atmospheric correction algorithm. Compared with Figure 1, when the SZA is large, the effective data products obtained by using this neural network atmospheric correction algorithm to process satellite images far exceed what can be typically obtained by the traditional atmospheric correction algorithms. Furthermore, the data are evenly distributed and do not contain spatial noise. Figure 9 shows the inversion results of neural network algorithm applied to other sea areas and different dates. As a comparison, the results of NIR algorithm are also shown in the figure. It can be seen that in the mid- and high-latitude sea areas in autumn and winter, the inversion effect of neural network algorithm is better, and there are obviously more effective data.

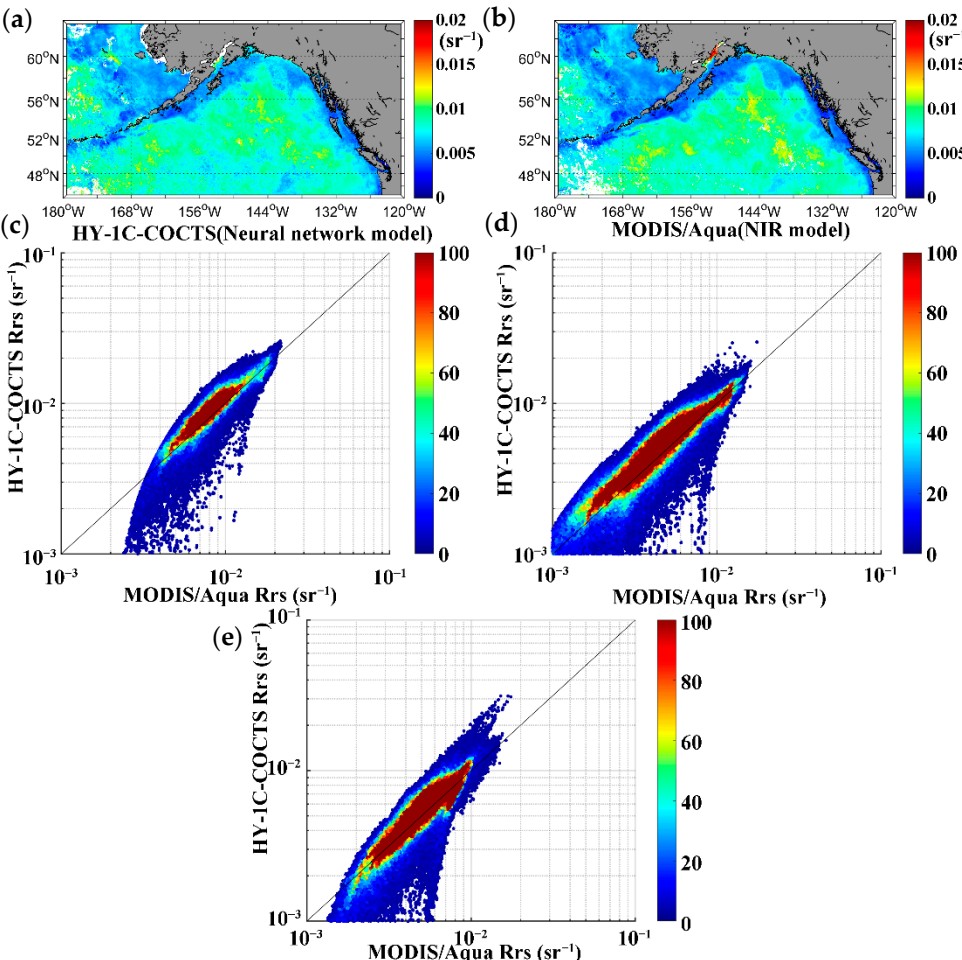

**Figure 7.** The monthly average Rrs product of HY-1C-COCTS and MODIS/Aqua in May 2020 calculated using the neural network algorithm and NIR iterative atmospheric correction algorithm. (**a**) The 443 nm Rrs product of HY-1C-COCTS. (**b**) The 443 nm Rrs product of MODIS/Aqua. (**c–e**) Comparison of the results of two algorithms for processing HY-1C-COCTS and MODIS/Aqua data in 412 nm, 443 nm, and 488 nm, respectively. The color bar represents the number of points.

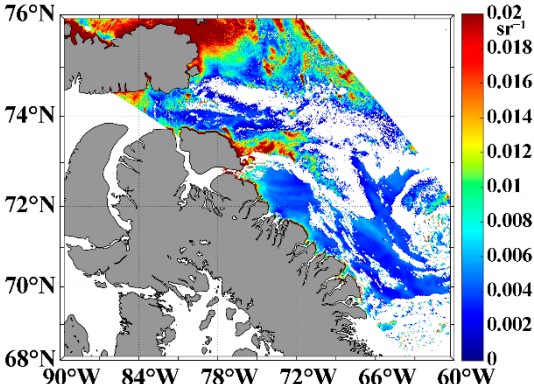

**Figure 8.** Rrs (443 nm) products from HY-1C-COCTS processed by the neural network atmospheric correction algorithm under a large SZA.

It can be seen from the satellite cross-comparison verification results that the atmospheric correction algorithm established for HY-1C-COCTS data with large SZAs is consistent with the results of the traditional atmospheric correction algorithm for data

with medium and low SZAs. Thus, the new algorithm can effectively process satellite data collected from environments where the SZAs were large.

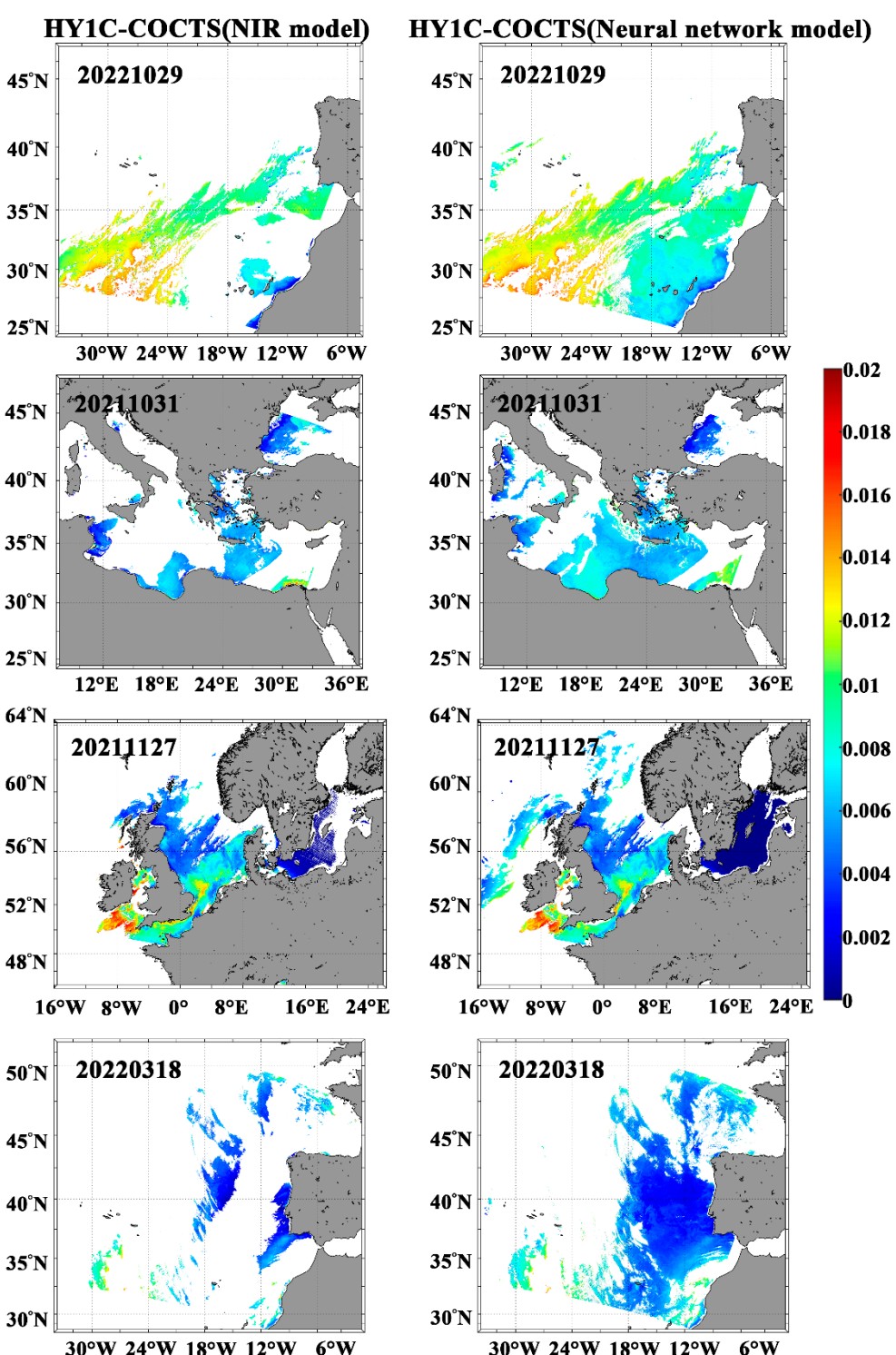

**Figure 9.** Rrs (443 nm) products from HY-1C-COCTS processed by the neural network atmospheric correction algorithm and NIR iterative atmospheric correction algorithm.

## 4. Discussion

After verifying the accuracy and stability of the algorithm, we used the algorithm to batch process the HY-1C-COCTS high latitude data. Figure 10 shows the HY-1C-COCTS monthly average Rrs product for October 2020 calculated using the neural network atmo-

spheric correction algorithm. When using the established algorithm to process satellite images, the effective data are sufficient, the processing efficiency is high, and the algorithm performance is stable. Therefore, this can be used to restore HY-1C-COCTS ocean color products when the SZA is large.

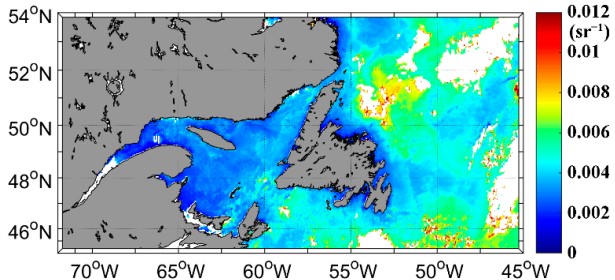

**Figure 10.** The monthly average Rrs product of HY-1C-COCTS in October 2020 calculated using the neural network algorithm.

Finally, we reconstructed the Rrs product dataset for the Arctic Ocean and surrounding high-latitude seas. This included the Rrs from the HY-1C-COCTS bands and the associated location information. Based on the restored Rrs, chlorophyll a and other marine ecological parameters can be further retrieved. The OC4O algorithm was used for chlorophyll concentration inversion, and the formula is as follows [33]:

$$\text{Chla} = 10^{\left(a_0 + \sum_{i}^{4} a_i \left(log_{10}\left(\frac{Rrs(\lambda_{blue})}{Rrs(\lambda_{green})}\right)\right)^i\right)} \tag{6}$$

where $a_0, a_1, a_2, a_3,$ and $a_4$ are constants.

Figure 11 shows the retrieved chlorophyll concentration. The newly established atmospheric correction algorithm can effectively process HY-1C-COCTS satellite data and can also generate effective data during the winter half of the year (September to March of the next year) when the Arctic SZA is large. The effective data in the winter half of the year can include 60° north latitude, where it is helpful to understand the biochemical process of the Arctic Ocean in the winter months.

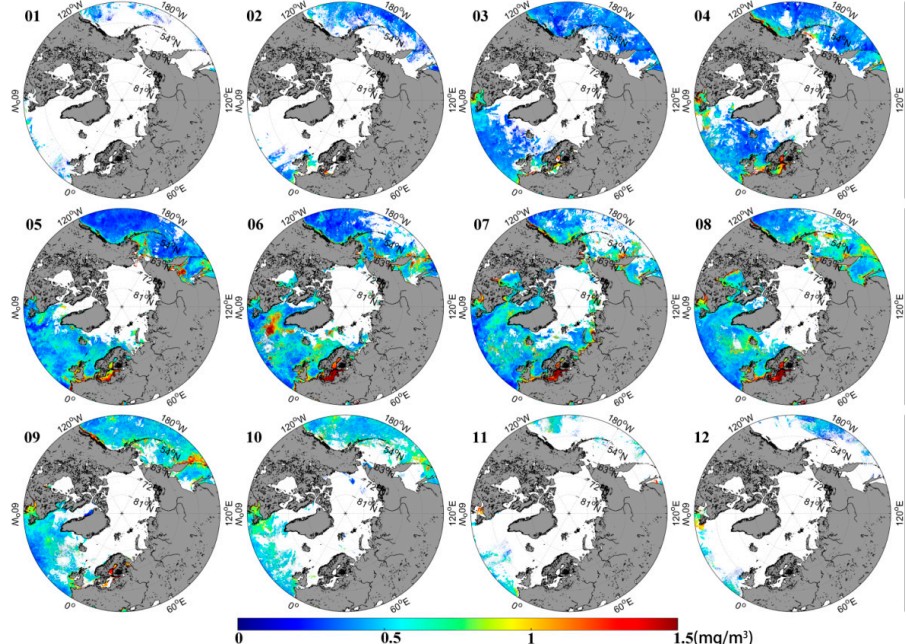

**Figure 11.** Chlorophyll concentration retrieved from the monthly average Rrs products of HY-1C-COCTS calculated by the neural network algorithm.

## 5. Conclusions

HY-1C-COCTS can provide daily ocean color data around the world. However, as a polar orbiting satellite, HY-1C will inevitably encounter large SZAs when observing areas in high-latitude seas. The current atmospheric correction model has a large error when dealing with observation data from regions with SZAs greater than 70°. To solve this problem, an atmospheric correction algorithm based on a neural network model was developed for HY-1C-COCTS data. The new algorithm used the multiple observation data from HY-1C-COCTS in the high-latitude region to determine the relationship between the Rrs with a small SZA and the Rayleigh-corrected radiance with a large SZA, thus establishing a direct relationship between the total radiance received by the satellite and the Rrs products. This provided a neural network training dataset that could be used to train the model to fit the observations, thus using the neural network to perform atmospheric correct for large SZAs. The evaluation results using satellite datasets showed that the inversion accuracy of the new algorithm is relatively high under different SZAs and different HY-1C-COCTS bands. The results of the cross-comparison with MODIS Aqua satellite products are largely consistent. After ensuring the accuracy and stability of the algorithm, we reconstructed the HY-1C-COCTS Rrs datasets of the Arctic Ocean and surrounding high-latitude oceans.

**Author Contributions:** Conceptualization, H.L.; data curation, H.L., J.D. and Y.B.; funding acquisition, X.H.; methodology, X.H. and Y.B.; project administration, D.W.; resources, J.D.; software, J.D., D.W., and T.L.; supervision, F.G. All authors have read and agreed to the published version of the manuscript.

**Funding:** This study was supported by the National Natural Science Foundation of China (Grants #41825014), the National Natural Science Foundation of China Youth Fund (Grants #42206183), the Scientific Research Fund of the Second Institute of Oceanography, MNR, grand no.SZ2222) and the Zhejiang Province Preferential Fund for Post Doctoral Research Projects (Grants #21636).

**Data Availability Statement:** Not applicable.

**Acknowledgments:** We thank the National Satellite Ocean Application Service for providing the observation data of HY-1C. We also thank NASA for providing MODIS/Aqua observation data. We thank the staffs of the satellite ground station, satellite data processing & sharing center, and marine satellite data online analysis platform of the State Key Laboratory of Satellite Ocean En-vironment Dynamics, Second Institute of Oceanography, Ministry of Natural Resources (SOED/SIO/MNR), for their help with the data processing.

**Conflicts of Interest:** The authors declare no conflict of interest.

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
