# Peer review of "The Inversion of HY-1C-COCTS Ocean Color Remote Sensing Products from High-Latitude Seas"

_remotesensing, doi:10.3390/rs14225722_

Round 1

Reviewer 1 Report

Dear authors and editors! I think that the paper is written on a relevant topic, but at the same time it is worth recognizing the strong limitation of using optical remote sensing data at high latitudes. In my opinion, section 3.3 should be deleted, since the article does not consider seasonal or interannual and variations in ocean color products. Section 3.3 should be replaced by more figures comparing the performance of the two algorithms (as is done in Figures 1 and 7).

p. 2, line 51. The term «Reflectance Factor» should be used instead of the term «RRS». See (Schaepman-Strub et al, About the importance of the definition of reflectance quantities-results of case studies, 2004), (chaepman-Strub et al,About the use of reflectance terminology in imaging spectroscopy, 2005).   p. 3, formulae (1). It is necessary to give an explanation of the physical meaning of Lra.

In the paper, it is necessary to give the accuracy of the MODIS L2 Rrs products and give the corresponding link.

Reviewer 2 Report

The authors have done a solid work on developing an atmospheric correction model for HY-1C-COCTS satellites with high SZAs. I recommend publishing it if authors can make some minor revisions.

 1.       Page 3, Line 87. The MODIS/Aqua Level-2 Rrs products from NASA were used for comparison and verification. The authors should give more introduction on these products, for example, how they were made and how their atmospheric corrections were made?

2.       What are the color bars in Figures 3, 4, and 6?

3.       Page 9, Line 260, “…NIR iterative atmospheric correction algorithm discussed in this paper.”  What is the NIR algorithm? Where it was discussed in this paper? Please clarify it.

4.       Page 11, Line 311, what’s the algorithm to retrieve CHL-a from the satellite images?

5.       ACOLITE is an atmospheric correction model for coastal and inland water applications. What’s the difference between your algorithm and ACOLITE?
